# BubbleSpec: Turning Long-Tail Bubbles into Speculative Rollout Drafts for Synchronous Reinforcement Learning

Yuhang Xu [* 1 2]   Kaibin Tian [* 2]   Yang Tian [2]   Zhice Yang [2]   Yifeng Yu [2]
Yan Li [† 2]   Shengzhong Liu [1]   Fan Wu [1]   Guihai Chen [1]

## Abstract

Reinforcement Learning (RL) has become a cornerstone for improving the performance of Large Language Models (LLMs). However, its rollout phase constitutes a significant efficiency bottleneck, mainly arising from the long-tail bubbles across data parallel ranks, particularly in long-context scenarios where faster GPUs remain idle while waiting for stragglers. Existing solutions, such as partial rollout or asynchronous RL, mitigate these bubbles by compromising the algorithm's strict synchronous nature. Instead, we propose **BubbleSpec**, a novel framework that accelerates RL rollouts while strictly keeping the mathematical exactness. Instead of attempting to eliminate bubbles, BubbleSpec exploits them. We exploit the idle time windows of faster ranks to pre-generate rollout results for subsequent steps, serving as drafts for speculative decoding. Unlike prior speculative methods that rely on historical epoch similarity and warm-ups, BubbleSpec is agnostic to dataset size and provides immediate acceleration from the onset of training. Extensive evaluations demonstrate that BubbleSpec reduces decoding steps by ∼**50%** and increases rollout throughput by up to **1.8×**. Critically, BubbleSpec is seamlessly compatible with various RL frameworks and strategies as it sustains the strict synchronous property of RL algorithms.

## 1. Introduction

The rapid evolution of Large Language Models (LLMs) has been significantly propelled by Reinforcement Learning

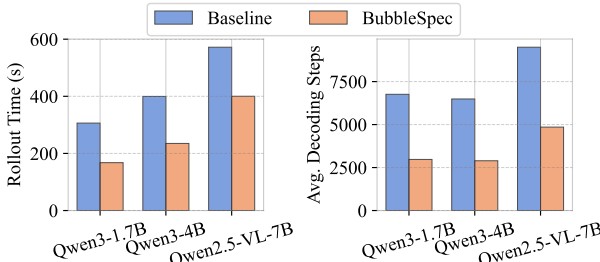

*Figure 1.* BubbleSpec reduces decoding steps by roughly 50% and delivers up to 1.8× speedup across different models.

(RL) (Ouyang et al., 2022; Kirk et al., 2023). The paradigm of "test-time scaling" has emerged as a frontier in enhancing reasoning capabilities, where models are incentivized to generate long Chain-of-Thought (CoT) (Wei et al., 2022; Lyu et al., 2023; Hu et al., 2026) reasoning paths to solve complex problems (Snell et al., 2024), represented by works like OpenAI-o1 (Jaech et al., 2024) and DeepSeek-R1 (Guo et al., 2025). In this context, RL plays a pivotal role in discovering and reinforcing these extended reasoning patterns, enabling models to achieve superior performance on mathematical and coding tasks through self-evolution.

However, the efficiency of RL training is largely constrained by the *rollout phase*, where the policy model samples responses for a batch of prompts. The stochastic nature of LLM generation causes unpredictable variance in response lengths, forcing faster devices to idle wait while waiting for stragglers—a phenomenon commonly known as "bubbles". To alleviate the rollout bubble issue, existing approaches (Team et al., 2025; Zhong et al., 2025a) relax the strict synchronization requirement between the rollout and policy update phases, allowing rollout workers to generate sequences continuously without frequent interruptions. However, studies (Xi et al., 2026; Qi et al., 2025; Liu et al.) have shown that such off-policy behavior and rollout–update discrepancies can lead to training instability.

Alternatively, speculative decoding, as a lossless LLM inference acceleration solution, has attracted increasing attention. In particular, model-free speculative decoding has been a leading option (He et al., 2025a; Liu et al., 2025a), owing to its lightweight draft overhead and the lack of sensitivity to

---
[*]Equal contribution [†]Project Leader [1]Shanghai Jiao Tong University [2]Bytedance. Correspondence to: Shengzhong Liu <shengzhong@sjtu.edu.cn>.

*Proceedings of the 43rd International Conference on Machine Learning*, Seoul, South Korea. PMLR 306, 2026. Copyright 2026 by the author(s).

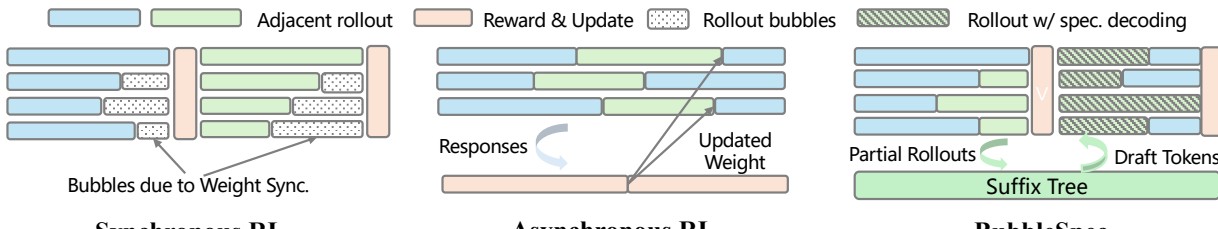

*Figure 2.* Compared to existing methods, BubbleSpec exploits rollout bubbles while maintaining the synchronous nature of RL.

rollout model weight evolution during RL updates. These methods typically exploit the strong similarity of responses across adjacent RL training epochs, reusing historical rollout outputs to construct draft candidates for the current batch, thereby accelerating rollouts while preserving the synchronous nature of RL training.

Nevertheless, methods based on cross-epoch response similarity face a fundamental limitation as the training dataset scales up, since they typically require an initial epoch to build a rollout cache for warm-up. In modern large-scale training regimes, such as those using DeepMath-103k (He et al., 2025b) and Polaris-53k (An et al., 2025), a single training epoch may span days to weeks. As a result, history-based approaches provide no acceleration during this prolonged initial stage (*i.e.*, the *cold-start problem*). Moreover, as the optimization steps within an epoch increase, the policy can evolve substantially, perturbing the distributional similarity between adjacent epochs and reducing the reusability of historical rollouts.

To overcome this limitation, we introduce **BubbleSpec**, a synchronous RL framework that fundamentally rethinks how idle time is managed in RL training. As shown in Figure 2, rather than eliminating bubbles at the cost of losing synchrony, BubbleSpec strategically *exploits* them through effective cross-step pipelining. During the idle window of the current rollout step, it proactively pre-generates rollouts for the next step and organizes the partial responses into a suffix tree that serves as a draft database for speculative decoding at the next step. Furthermore, we carefully design our speculative decoding technique, combining it with operator-level optimizations to minimize both draft and verification costs, thereby achieving high end-to-end efficiency for large-batch RL rollouts, a regime where conventional speculative decoding typically fails to provide acceleration.

We implement BubbleSpec based on the Verl framework (Sheng et al., 2025), and evaluate it in long-context RL scenarios. Experimental results show that our BubbleSpec reduces decoding steps by more than 50% and improves rollout throughput by up to 1.8×. Compared to prior arts, BubbleSpec offers distinct advantages: (1) **Immediate Acceleration**: It provides speedup from the very first training step, making it uniquely suitable for large-scale dataset training where epoch-based history is unavailable or stale. (2)

**Strict Synchronous Guarantee**: By adhering to rigorous speculative decoding protocols, BubbleSpec ensures the rollout distribution remains mathematically identical to the original policy. This allows it to be seamlessly applied to various RL algorithms (*e.g.*, GSPO, DAPO, SAPO (Gao et al., 2025a; Yu et al., 2025; Zheng et al., 2025)) without requiring algorithmic modifications or risking performance degradation. In summary, our contributions can be summarized as follows:

- We deeply analyze long-tail effects in synchronous RL and highlight key considerations for applying speculative decoding in RL training.

- We propose BubbleSpec, the first synchronous RL framework that converts long-tail bubbles into speculative drafts, overcoming the cold-start limitation of history-based approaches.

- We design a deployment-oriented speculative decoding scheme that ensures reductions in decoding steps translate into end-to-end efficiency gains.

- Extensive evaluations on realistic long-context RL scenarios demonstrate that BubbleSpec consistently enhances rollout efficiency across various model sizes.

**Conflict of Interest Disclosure.** The authors declare that they have no financial conflicts of interest related to this work.

## 2. Background and Motivation

### 2.1. Long-Tail Effect in RL Rollouts

RL algorithms such as PPO (Schulman et al., 2017) and GRPO (Guo et al., 2025) typically follow a synchronized iterative cycle: (1) *rollout*, (2) *reward calculation*, and (3) *actor update*. Among these, the rollout phase is most time-consuming, often accounting for over 70% of total training time and thus becoming the primary bottleneck in RL training (He et al., 2025a; Hu et al., 2025).

In the rollout phase, the policy model usually samples multiple responses for a batch of prompts across distributed data-parallel (DP) ranks. A key challenge arises from the inherent variance in generation length, due to both the stochastic

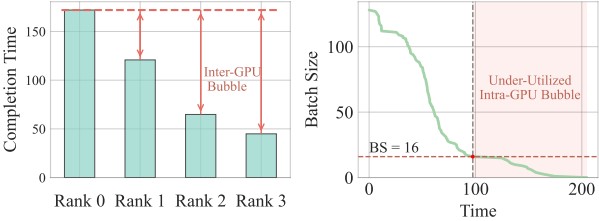

*Figure 3.* Inter-GPU and intra-GPU bubbles in RL rollouts

nature of sampling and prompt diversity. This variability induces severe long-tail effects, especially in long-context RL training, leading to GPU idleness (called *bubbles*). More specifically, these bubbles can be divided into two categories: (1) **Inter-GPU Bubbles.** DP ranks that finish all assigned prompts must wait for the slowest rank to complete, leaving some GPUs completely idle, as illustrated in Figure 3 left part. (2) **Intra-GPU Bubbles.** Within a DP rank, most generations have relatively short responses and terminate early, so the remaining computation proceeds with a much smaller effective batch size. In this case, GPUs are not fully idle, but their utilization is low, as shown in the right part of Figure 3. These bubbles can account for more than 60% of the total rollout time, making the rollout phase GPU-inefficient and difficult to scale.

To mitigate these bubbles, prior work has explored relaxing synchronization constraints. Techniques such as asynchronous RL (Fu et al., 2025) decouple rollout workers from the actor update, enabling continuous response generation with minimal synchronization overhead for weight updates. However, these methods inevitably introduce response staleness, which can destabilize RL training, as increasingly noted in recent studies (Xi et al., 2026; Qi et al., 2025; Liu et al.). Given the crucial importance of maintaining synchrony, our work instead aims to accelerate the rollout phase by leveraging these inevitable bubbles without compromising the synchronous property.

▶ **Insight:** Rather than eliminating bubbles, BubbleSpec proactively exploits them to pre-generate rollouts, forming **cross-batch pipelining for RL training**. To avoid the distributional mismatch between generation and training, BubbleSpec uses these responses for suffix-tree-based speculative decoding to accelerate LLM inference.

### 2.2. Model-Free Speculative Decoding

Speculative decoding (Leviathan et al., 2023) accelerates LLM inference via a draft-then-verify paradigm: it first generates draft tokens using a lightweight method, then verifies them in parallel with the target model. Crucially, rejection sampling ensures that the final output distribution is mathematically identical to standard auto-regressive sampling (Leviathan et al., 2023).

While speculative decoding greatly reduces the number of forward calls of the target model, *i.e.*, decode steps, the per-step latency may increase due to two factors: (1) **Draft overhead**, the additional time needed to generate draft tokens; and (2) **Verification overhead**, the increased target-model forward latency caused by a larger and varying-length batch. Assuming the decoding-step reduction ratio and per-step latency increase ratio are $\alpha$ and $\mu$, respectively, the end-to-end generation time reduction ratio is

$$\text{Speedup} = 1 - (1 - \alpha)(1 + \mu). \tag{1}$$

Model-based speculative methods incur substantial draft overhead, causing their acceleration to deteriorate at the large batch sizes typical of RL rollouts (Li et al., 2025). In contrast, model-free approaches typically generate draft tokens via prefix pattern matching within existing sequences, achieving much lower draft overhead. Consequently, recent work has explored model-free approaches to accelerate RL rollouts (He et al., 2025a; Liu et al., 2025a). These methods typically exploit inter-epoch similarity by reusing rollouts from the previous epoch as drafts for the current epoch. While promising, they are mainly validated with training on small-scale datasets and exhibit two major limitations. First, they provide **no acceleration during the initial epoch**, which is a critical bottleneck in large-scale training where a single epoch may span hundreds or thousands of steps and last days or even weeks. Second, as the dataset size and the steps per epoch increase, the **policy model can change drastically between epochs**. This divergence reduces the relevance of historical rollouts, degrading draft quality and diminishing speculative speedup.

▶ **Insight:** Our key intuition is to **leverage per-step rollout bubbles to generate drafts for subsequent steps**, while keeping draft and verification overhead low, thereby achieving consistent speedup throughout the training process.

## 3. BubbleSpec Design

BubbleSpec provides a systematic, lossless rollout-phase optimization that leverages idle bubbles to efficiently pre-generate draft responses and thereby enable fast speculative decoding, while remaining a modular component that can be seamlessly integrated into various synchronous RL frameworks. In subsection 3.1, we describe how BubbleSpec schedules rollout pre-generation into these idle bubbles; we then show how the resulting partial rollouts are used for efficient speculative decoding with low draft overhead (subsection 3.2) and verification overhead (subsection 3.3), ensuring that the reduction in decoding steps translates directly into a reduction in rollout time.

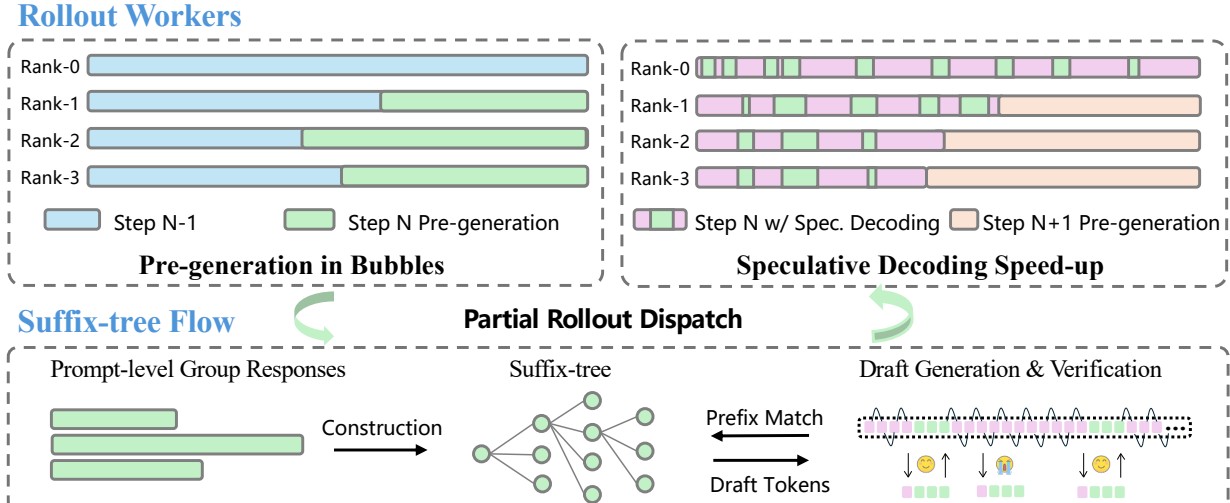

Figure 4. Overview of BubbleSpec. We illustrate the workflow across three consecutive training steps; for simplicity, we only show the speculative decoding process for step $N$.

### 3.1. Use GPU Bubbles for Rollout Pre-Generation

As mentioned, our key objective is to harvest idle compute for draft pre-generation while avoiding interference with the main rollout and keeping synchronization overhead low.

**Inter-GPU Bubbles vs. Intra-GPU Bubbles.** We first consider how to utilize bubbles without disrupting current batch generation, and in particular, whether to exploit inter-GPU or intra-GPU bubbles. Intra-GPU bubbles can expose more time for pre-generation, but they directly contend with the running decoding workload. We experiment with a selective intra-GPU strategy: among all DP ranks, we use only the intra-GPU bubbles of the fastest ranks (*e.g.*, 6 out of 8) once their effective batch size drops below a threshold. We observed highly unstable interference: Fast ranks are slowed down after taking additional requests and may even finish later than the original stragglers. Instead, we found inter-GPU bubbles alone are sufficient to generate enough draft tokens while avoiding extra contention and instability. Therefore, we exploit only the inter-GPU bubbles for rollout pre-generation.

**Rollout DP Rank Synchronization.** Another key design choice is how to synchronize rollout DP ranks so that pre-generation can stop once the slowest rank finishes the current batch. Since per-step decoding lasts for only a few milliseconds, naive cross-GPU or cross-node synchronization can incur disproportionate overhead. We therefore adopt a periodic polling scheme: during the pre-generation of $B_{t+1}$, each rank queries a central synchronizer every $T$ decoding steps. If the synchronizer reports that all ranks have completed $B_t$, pre-generation is halted, and the system proceeds to the barrier. The parameter $T$ is tuned to balance the freshness of synchronization against the cost of inter-process communication, keeping coordination overhead negligible

compared to decoding.

**Pre-Generation Batch Size.** Finally, we decide how many samples to pre-generate per prompt. A larger pre-generation batch size explores more decoding paths for a single prompt, increasing diversity and the probability of prefix matches during speculative decoding. However, an excessively large batch size also increases per-step decoding latency, which can reduce the maximum response length that can be served and may leave prompts with long responses under-covered at later positions. We empirically match the number of pre-generated samples per prompt with the number of samples used in GRPO for each prompt group, and find that this choice well balances the diversity and coverage.

### 3.2. Suffix Tree-based Speculative Decoding

To exploit pre-generated partial rollouts as speculative drafts, the system should repetitively match the current rollout prefix against a large pool of historical tokens and retrieve likely continuations with minimal online overhead, *i.e.*, draft overhead. This requirement naturally leads to a design that separates *offline indexing* from *online lookup*.

We consider two generic indexing strategies for prefix matching over historical rollouts: 1) an *n-gram–style* scheme that performs pattern matching directly over raw token sequences, and 2) a *suffix-based* scheme that builds an explicit index over all suffixes. The former requires no explicit index, but its per-query cost is proportional to the history length, which can be prohibitive over thousands of decoding steps. The latter amortizes work by building a compact suffix index over the pre-generated tokens, so that lookup depends primarily on the prefix length and the number of matches, and is effectively decoupled from the total history size.

Table 1. Latency of attention operators in speculative decoding.

| Normal Attention w/o. Speculative Queries | Batch Split Attention | | Unified Attention |
|---|---|---|---|
| | Prefill | Decode | |
| 0.372 ms | 0.753 ms | 0.226 ms | 0.380 ms |

Since speculative decoding issues a large number of prefix-matching queries during rollout, we adopt the **suffix-based design**. Conceptually, the system aggregates pre-generated rollouts into prompt-associated token pools and constructs a suffix index for each pool. The indices are sharded across rollout workers following the existing rollout dispatch: each rollout DP rank builds and maintains suffix indices only for the prompts it is responsible for generating. As a result, the construction cost depends on the local prompt set instead of the global training scale, and the indexing layer scales with system size without a centralized bottleneck. This parallelization bounds the suffix tree construction overhead as the system scales up.

At each decoding step, we retrieve a block of candidate tokens conditioned on the current prefix, which provides a deterministic proposal for model-free speculative decoding. Concretely, let $\tilde{x}_t$ denote the retrieved draft token at step $t$, and let $p_t(\cdot)$ be the *actual* decoding distribution used by the target policy at that step, after temperature scaling and any top-$p$/top-$k$ filtering. We accept $\tilde{x}_t$ with probability

$$P_{\text{accept}} = p_t(\tilde{x}_t). \tag{2}$$

Equivalently, this is rejection sampling with a degenerate proposal $q_t(\tilde{x}_t) = 1$. If the draft token is rejected, we sample a recovered token from the residual distribution

$$r_t(x) = \frac{p_t(x)\mathbf{1}[x \neq \tilde{x}_t]}{1 - p_t(\tilde{x}_t)}. \tag{3}$$

In the multi-token case, we verify a retrieved draft block sequentially and accept tokens until the first rejection; upon rejection, we draw one recovered token from the corresponding residual distribution and restart drafting from the updated prefix. If all draft tokens are accepted, we continue decoding from the extended prefix. Therefore, BubbleSpec exactly preserves the target rollout distribution while reducing the number of decoding steps. Compared with history-based methods that rely on cross-epoch policy similarity, our acceptance behavior depends on the quality of the suffix-tree proposal under the current policy, while the correctness guarantee follows directly from the rejection-sampling construction. A full pseudocode of the speculative decoding procedure is provided in Appendix A.2.

### 3.3. Unified Attention for Speculative Decoding

As analyzed in subsection 2.2, although suffix decoding reduces decoding steps with low draft overhead, its actual acceleration is constrained by increased per-step latency

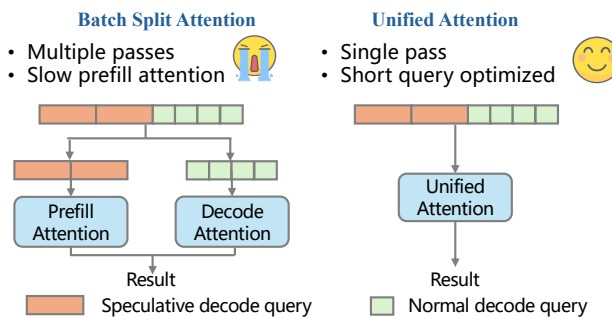

Figure 5. Comparison between batch split and unified attention.

due to the verification overhead. We find that even when the number of draft tokens is limited (*i.e.*, introducing little extra computation and keeping the system memory-bound), there is still a substantial increase in per-step decoding latency, offsetting the benefit of fewer steps. While prior work typically attributes this inefficiency to the increased computation from larger batch sizes, we identify that it is mainly caused by *suboptimal batching behavior with draft tokens*.

Operations in the LLM forward pass can be divided into: (1) **Token-wise** operations, such as linear layers and layer normalization, which can be efficiently batched regardless of request length; and (2) **Request-wise** operations, primarily attention, whose performance is highly sensitive to batching strategies. The draft tokens make the attention query length greater than 1, similar to prefill attention. Existing inference engines use different kernels for prefill and decode attention due to their distinct resource characteristics. When draft tokens appear in speculative decoding, they trigger the less efficient prefill-style attention. As shown in Figure 5, this split-attention behavior partitions the batch into two groups: requests with draft tokens use prefill attention, and other requests use standard decode attention.

We show this inefficiency in Table 1. The experiment is conducted with batch size 128 and context length 8k, where 32 requests are speculative with 4 draft tokens each. Although the total number of tokens increases only from 128 to 256, the latency under batch-split attention is nearly three times that of normal attention without draft tokens, primarily due to the prefill attention.

To address these issues, we optimize the attention implementation within the rollout engine by adopting a unified attention operator capable of handling variable query lengths within a short range, as shown in Figure 5. This unified operator processes normal and decode queries within a single CUDA kernel launch, eliminating batch splitting overhead. The matrix multiplications for short-query verification are efficiently executed on tensor cores (Markidis et al., 2018) in modern GPUs, whose fixed tile sizes ensure negligible latency increase as long as the problem size remains small. As shown in Table 1, unified attention incurs almost no additional latency for speculative requests compared to nor-

mal attention without speculative queries, ensuring that the reduced number of decoding steps translates into real end-to-end generation speedup.

# 4. Experiments

## 4.1. Experimental Setup

**Models and datasets.** We evaluate BubbleSpec on the Qwen model series, including Qwen3-1.7B, Qwen3-4B (Yang et al., 2025), and Qwen2.5-VL-7B (Bai et al., 2025). For Qwen3-1.7B and Qwen3-4B, we perform cold-start RL training from scratch. For Qwen2.5-VL-7B, training is initialized from a checkpoint that has been fine-tuned to enhance long chain-of-thought (CoT) reasoning capabilities. All models are trained using samples from Polaris-53k (An et al., 2025), DeepMath-103K (He et al., 2025b), and SimpeRL-Zoo-Data (Zeng et al., 2025). All experiments are conducted on a single node equipped with 8 NVIDIA GPUs.

**Configurations and hyperparameters.** For RL algorithms, we adopt SAPO (Gao et al., 2025a), as it demonstrates superior stability in long-context RL training. Notably, BubbleSpec does not require any algorithmic modifications and can be directly applied to other algorithms such as GRPO and DAPO, since they share the same rollout strategy but differ only in advantage estimation or loss computation. The batch size is set to 64, and we sample 16 responses for each prompt, resulting in a total rollout batch size of 1024. Additionally, we sample 16 responses for draft pre-generation of the next batch. The rollout temperature is set to 1.0. For Qwen3-1.7B and Qwen3-4B, the maximum response length is set to 48K, while for Qwen2.5-VL-7B, the maximum response length is set to 64K. For suffix speculative decoding, we use a fixed draft length of 4 tokens. Synchronization across rollout data-parallel ranks is performed every 50 decoding steps.

**Metrics.** We use the rollout time, *i.e.*, the end-to-end completion time of the latest rollout DP rank, to quantify efficiency. For speculative decoding, we measure the average and maximum decoding steps, along with the response length, to assess the step reduction achieved by speculative decoding. Besides, we report the acceptance length, draft length, and acceptance rate for speculative decoding. All metrics are averages across 50 training steps.

## 4.2. Main Results

**Rollout Efficiency.** We present the main results in Table 2. Across models of varying sizes, BubbleSpec consistently reduces decoding steps by approximately half compared to vanilla Verl, with average decoding steps reduced by 48.9% to 56.8% and maximum decoding steps reduced by 43.8% to 59.6%. Rollout time is decreased by 30.0% to 45.2%,

leading to a throughput improvement of $1.4\times$ to $1.8\times$. The maximum decoding step is closely coupled with rollout time, as the overall rollout time is determined by the slowest DP rank. Notably, the proportionate reduction in rollout time is less than that of decoding steps, primarily because speculative decoding operates on larger batch sizes, incurring additional overhead. We also observe that speedup slightly diminishes with increasing model size. This is partially because larger models exhibit higher computational intensity, causing large batch sizes to more readily shift the workload into computation-bound regimes, thereby offsetting the benefits of speculative decoding.

We also report results of the speculative decoding metrics. The average acceptance length is $\sim 2$ tokens (including the recovered token and bonus token). We observe that the acceptance rate for model-free speculative decoding decreases quickly for later tokens. Specifically, for Qwen3-1.7B, the distribution of accepted streak lengths from 1 to 4 is 14.2%, 58.6%, 25.5%, and 1.7%, respectively. Empirically, using 4 draft tokens strikes a good balance between the number of verified tokens and the additional overhead.

**Details of next-step draft pre-generation.** We provide details on rollout pre-generation for next steps in Table 3. The average and maximum rollout bubble times are measured across all rollout DP ranks. On average, bubble time accounts for over 1/3 of the total rollout time, and the maximum bubble time can exceed 2/3, indicating severe load imbalance and significant GPU idleness across ranks. These bubbles provide ample slack for draft pre-generation in the next step, allowing partial rollouts to reach up to 2/3 of the full response length, thereby ensuring a high matching rate for speculative decoding in the subsequent step.

Additionally, the overhead of suffix-tree construction is reported in Table 4. This overhead is negligible—less than 1% of the rollout time. The construction runs in linear time with respect to the token numbers. We dispatch pre-generated draft responses to each rollout rank according to their assigned prompts and build suffix trees in parallel, keeping the overhead low even as the number of ranks scales up. Moreover, since suffix-tree construction uses CPU resources, it can be further optimized by running in a separate background process, with its latency fully hidden behind reward computation and actor update.

**Accuracy integrity.** We report the accuracy on Qwen2.5-VL-7B-Instruct after post-training with BubbleSpec and Verl in Table 5. Both RL runs start from the same checkpoint obtained after SFT on the AceReason (Liu et al., 2025b) dataset. BubbleSpec and Verl achieve comparable performance on both the in-domain text test set and the out-of-domain vision test set, demonstrating the losslessness of speculative decoding and the preservation of synchronous training.

*Table 2.* Main results on rollout efficiency and speculative decoding.

| Method | Rollout Time (s) | Decoding Steps | | Response Length | | Speculative Metrics | | |
|---|---|---|---|---|---|---|---|---|
| | | Average | Maximum | Average | Maximum | Accept. Length | Draft Length | Accept. Rate |
| **Qwen3-1.7B** | | | | | | | | |
| Verl | 306.2 | 6741 | 39531 | 6741 | 39361 | N/A | N/A | N/A |
| BubbleSpec | 167.6 (**-45.2%**) | 2913 (**-56.8%**) | 15908 (**-59.6%**) | 6592 | 39685 | 2.15 | 3.84 | 29.94% |
| **Qwen3-4B** | | | | | | | | |
| Verl | 399.6 | 6489 | 37910 | 6481 | 37910 | N/A | N/A | N/A |
| BubbleSpec | 235.0 (**-41.1%**) | 2895 (**-56.1%**) | 17616 (**-53.5%**) | 6499 | 38577 | 2.09 | 3.81 | 28.6% |
| **Qwen2.5-VL-7B** | | | | | | | | |
| Verl | 571.9 | 9500 | 60509 | 9500 | 60509 | N/A | N/A | N/A |
| BubbleSpec | 400.1 (**-30.0%**) | 4853 (**-48.9%**) | 34027 (**-43.8%**) | 9414 | 59912 | 1.81 | 3.73 | 21.7% |

*Table 3.* Details on next-step draft pre-generation.

| Model | Rollout Bubble Time (s) | | Draft Response Length | |
|---|---|---|---|---|
| | Average | Maximum | Average | Maximum |
| Qwen3-1.7B | 61.9 | 109.1 | 4169 | 24029 |
| Qwen3-4B | 91.1 | 164.7 | 4434 | 25188 |
| Qwen2.5-VL-7B | 157.7 | 283.2 | 6802 | 46824 |

*Table 4.* Overhead of suffix tree construction.

| Model | Qwen3-1.7B | Qwen3-4B | Qwen2.5-VL-7B |
|---|---|---|---|
| Average Overhead | 1.2s | 1.2s | 2.0s |
| Maximum Overhead | 2.5s | 2.0s | 2.8s |

*Table 5.* Accuracy after RL training by BubbleSpec and Verl. Both BubbleSpec and Verl continue training after the SFT stage.

| Model | Text (In-Domain) | | | | Vision (OOD) |
|---|---|---|---|---|---|
| | AIME24 | AIME25 | MATH500 | GSM8K | Geo3K |
| Qwen2.5-VL-7B | 3.86 | 0.83 | 67.2 | 85.97 | 38.94 |
| +SFT | 76.77 | 59.89 | 95.4 | 91.74 | 47.92 |
| +BubbleSpec | 79.58 | 62.81 | 95.6 | 92.34 | 51.41 |
| +Verl | 80.72 | 63.43 | 95.8 | 91.96 | 48.25 |

*Table 6.* Ablation study on unified attention using Qwen3-1.7B.

| Method | Rollout time | Dec. Time/Step |
|---|---|---|
| BubbleSpec | 167.6s | 10.48ms |
| BubbleSpec w/o. unified attention | 294.9s | 18.44ms |

*Table 7.* Comparison between suffix decoding and n-gram decoding on Qwen3-1.7B

| Method | Rollout Time (s) | Accept. Length | Decoding Steps | |
|---|---|---|---|---|
| | | | Max. | Avg. |
| Suffix Decoding | 167.6 | 2.15 | 15908 | 2913 |
| N-gram Decoding | 305.9 | 1.83 | 24269 | 3854 |

only verification incurs measurable overhead. During speculative verification, linear layers for requests with varying query lengths can be efficiently batched, whereas attention operations typically rely on multiple kernels optimized for different query and context lengths. Unified attention avoids unnecessary kernel launches and is tuned for the short query lengths characteristic of speculative decoding, thereby delivering superior performance.

**Suffix decoding vs. n-gram decoding.** BubbleSpec adopts a suffix tree for pattern matching against pre-generated responses. Under the same settings, we also evaluated the n-gram method for pattern matching within BubbleSpec, with results presented in Table 7. We make the following observations: First, the acceptance length of suffix decoding is higher than that of n-gram decoding. This is because suffix decoding selects candidate tokens with higher confidence based on token node occurrence frequencies, whereas n-gram decoding merely performs a linear match of repeated token sequences to return the candidate with the longest common prefix. Second, although n-gram decoding achieves a considerable reduction in decoding steps, this does not translate to a reduction in overall rollout time. This is attributed to the computational overhead of n-gram prefix matching; the time complexity grows linearly with the length of historical responses, thereby offsetting the benefits gained from fewer decoding steps.

**Sensitivity of pre-generation sample count.** As shown in Table 8, we report both rollout efficiency and draft pre-generation metrics. In general, a larger pre-generation batch size explores more decoding paths for a single prompt, increasing diversity and thus the probability of prefix matches during speculative decoding. However, an excessively large batch size also increases per-step decoding latency, which

## 4.3. Extended Studies

**Contribution of unified speculative attention.** To isolate the impact of the unified attention, we conduct an ablation on Qwen3-1.7B, as shown in Table 6. All settings are identical except that one variant employs unified attention for speculative decoding, while the other uses batch-split attention. We observe that the average decoding-step latency under batch-split attention nearly doubles, substantially offsetting the gains from fewer decoding steps achieved via speculative decoding. Sustaining speculative decoding efficiency at large batch sizes has long been challenging; for example, EAGLE-3 (Li et al., 2025) reports no improvement or even degradation under such regimes. A key advantage of the suffix-based speculative decoding is that draft-generation overhead is negligible compared to model-based methods;

*Table 8.* Ablation study on the number of samples per pre-generated prompt.

| Samples Per Prompt | Rollout Time(s) | Max. Dec. Steps | Draft Response Length | |
|---|---|---|---|---|
| | | | Avg. | Max. |
| 4 | 183.8 | 17455 | 6889 | 27880 |
| 16 | 167.6 | 15908 | 4169 | 24029 |
| 32 | 184.4 | 18028 | 2960 | 18028 |

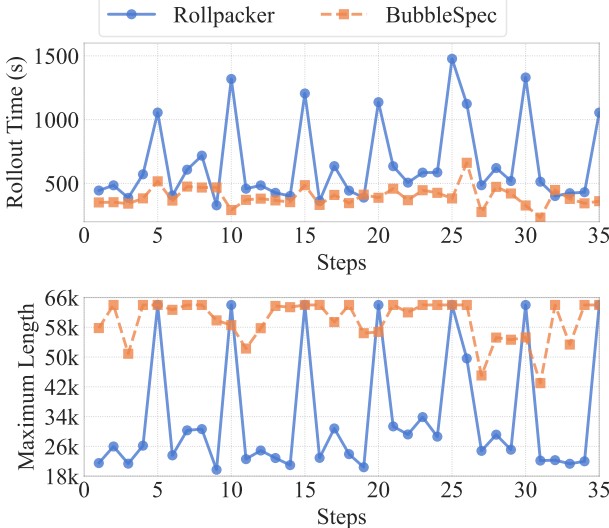

*Figure 6.* Comparison of BubbleSpec and RollPacker on Qwen2.5-VL-7B in terms of maximum response length and rollout time.

reduces the maximum response length that can be served and may cause prompts with longer responses to have insufficient draft coverage at later positions. Overall, we find that generating 16 responses per pre-generated prompt achieves a good balance between generation diversity and draft response length.

### 4.4. Compared with RollPacker's Batch Reordering

RollPacker (Gao et al., 2025b) aims to mitigate rollout bubbles while preserving the synchronous nature of RL. To achieve this, it employs a *tail batching* technique. Specifically, for a target batch size $B$, RollPacker initiates the generation process with an expanded set of $\eta B$ prompts (where $\eta > 1$). The current batch generation terminates once $B$ prompts have completed generation. The remaining unfinished "tail" prompts are stored in a queue; once the queue size reaches $B$, these prompts are retrieved and processed in a dedicated tail round. We implemented Roll-Packer within Verl, adhering to the authors' recommended setting of $\eta = 1.25$ (*i.e.*, generating 80 prompts per step with 20 samples per prompt), which corresponds to executing one tail round every 5 steps.

We present a comparison of rollout time and maximum sequence length on Qwen2.5-VL-7B in Figure 6. It can be observed that although the maximum response length in

RollPacker's short rounds is significantly shorter, Bubble-Spec still achieves lower rollout times for the majority of steps. This is primarily due to: (1) the reduction in decoding steps enabled by speculative decoding, and (2) the additional computational overhead introduced by RollPacker's larger batch size. Furthermore, RollPacker's rollout time in tail rounds far exceeds that of BubbleSpec, resulting in a significantly higher average rollout time of 655 s compared to BubbleSpec's 397 s. We observe that RollPacker's performance gains depend heavily on the sparsity of long responses; consequently, its efficiency degrades as the disparity in maximum response length between short and tail rounds diminishes. In contrast, although BubbleSpec similarly relies on long-tail requests to enable pre-generation, it maintains consistent performance gains even when bubble capacity is limited.

## 5. Related Works

**Speculative decoding in RL.** Speculative decoding is a widely adopted for accelerating LLM inference. It mitigates the auto-regressive generation inefficiency by rapidly drafting candidate tokens and verifying them in parallel. Through rejection sampling, it ensures an output distribution identical to that of the target model (Leviathan et al., 2023). Existing methods can be categorized into two streams. *Model-based approaches* employ a lightweight module to predict multiple draft tokens efficiently (Li et al., 2024; 2025; Cai et al., 2024; Ankner et al., 2024; Yi et al., 2024), exemplified by methods such as EAGLE and Medusa. Conversely, *model-free approaches* such as suffix-decoding typically generate drafts via token pattern matching within existing sequences (Oliaro et al., 2025; Luo et al., 2025; Saxena, 2023). Leveraging the lossless property of speculative decoding, recent research has explored its potential to accelerate RL rollout. For instance, Rhyme-RL (He et al., 2025a) and SpecRL (Liu et al., 2025a) utilize historical rollout results for draft generation. However, they require an initial warm-up epoch, rendering them less suitable for RL training on large-scale datasets. TLT (Hu et al., 2025) pioneers model-based speculative decoding in RL, addressing the policy evolution by continuously training the draft model. While promising, it incurs the overhead of managing a separate draft model. This is particularly problematic in modern pipelines where RL training typically follows a SFT stage, meaning a compatible draft model is not directly available.

**Efficient RL Systems.** Much of the existing research (Sheng et al., 2025; Zhong et al., 2025b; Hu et al., 2024; Chen et al., 2026; 2025) has concentrated on improving LLM decoding efficiency and managing the complex RL workflows, aiming to maximize training throughput and GPU utilization across heterogeneous RL components. Subsequent studies have pinpointed the rollout process as the

principal bottleneck in long-context RL training. This limitation arises primarily from the memory-bound nature of LLM token generation and imbalanced workloads across data-parallel ranks. Several works (Zhu et al., 2025; Zhong et al., 2025a; Fu et al., 2025) tackle this inefficiency by relaxing synchronization requirements in RL training. Areal (Fu et al., 2025) proposes partial rollout, wherein generation continues from prefixes produced by a previous policy model. StreamRL (Zhong et al., 2025a) alleviates synchronization constraints by permitting some staleness in rollout samples. While these methods enhance throughput, they may introduce unpredictable performance degradation. As recent studies (Xi et al., 2026; Qi et al., 2025; Liu et al.) emphasize, off-policy training and training-inference mismatches can negatively impact RL stability. RollPacker (Gao et al., 2025b) mitigates rollout bubbles by batch reordering, allowing long rollouts to be processed in a single extended round. This approach avoids synchronization issues; however, our evaluation reveals that its effectiveness is highly contingent on the sparsity of samples with long responses.

## 6. Conclusion

We presented BubbleSpec, a framework designed to transform the inefficiency of rollout bubbles into a valuable computational resource for large-scale RL training. Unlike prior works that rely on relaxed synchronization or necessitate warm-up, BubbleSpec proactively leverages idle GPU time for suffix-driven speculative decoding, providing consistent speedup throughout training. Empirical results validate that BubbleSpec significantly reduces decoding steps and boosts throughput. Crucially, this acceleration is achieved without compromising the mathematical equivalence of the generation process, ensuring the RL training stability and model performance. BubbleSpec offers a robust, plug-and-play solution for accelerating long-context reasoning models. We hope this work inspires further research into utilizing idle computational cycles in synchronous training paradigms.

## Acknowledgements

This work was sponsored in part by the Fundamental and Interdisciplinary Disciplines Breakthrough Plan of the Ministry of Education of China (No. JYB2025XDXM103), in part by China NSF grant No. 62472278, 62432007, 62441236, 62332014, 62332013, and 62225202, and Shanghai QiYuan Innovation Foundation. This work was partially supported by SJTU Kunpeng & Ascend Center of Excellence. The opinions, findings, conclusions, and recommendations in this paper are those of the authors and do not necessarily reflect the views of the funding agencies or the government.

## Impact Statement

*BubbleSpec* improves the efficiency of synchronous RL training for large language models by reducing idle time during rollout, which can shorten experimentation cycles and lower the per-run cost of post-training. These benefits may help researchers make better use of limited compute resources, especially in long-context settings. At the same time, making RL training cheaper and faster may also accelerate the scaling and deployment of increasingly capable models, including in settings where such models could be misused. Moreover, lower per-run cost does not necessarily reduce overall resource consumption, since improved efficiency can encourage more training at larger scales, and such benefits may disproportionately favor organizations with access to large distributed infrastructure. As a systems optimization method, *BubbleSpec* does not directly introduce new model capabilities, and its broader societal impact will depend on how it is used, evaluated, and deployed in practice.

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

## A. Evaluation Details and Additional Results

### A.1. Experimental Details

BubbleSpec is implemented on top of `Verl-0.5.0.dev0`, using `vLLM-v0.10.1` as the rollout backend. The tensor-parallel size is set to 1. In general, smaller tensor-parallel sizes reduce inter-GPU communication overhead, but also decrease the available GPU memory for the KV cache, which can trigger runtime KV-cache swapping and request preemption. In such cases, long-tail requests may require more decoding steps. We therefore recommend choosing the smallest tensor-parallel size that does not incur severe KV-cache swapping.

For cross data-parallel rank synchronization, we use a Ray actor as a centralized coordinator. The unified attention implementation adopts TensorRT operators from FlashInfer, and the suffix-tree decoding is based on Arctic-Inference. We set the GPU memory utilization to 0.7 and enable both the chunked prefill and CUDA graph modes of vLLM. The maximum number of batched tokens for rollout is set to 64k. Rollout sampling uses temperature 1.0 and top_p 1.0.

For policy updates, we adopt the AdamW optimizer with a learning rate of $1e-6$ and weight decay of 0.01. For all samples, we use a rule-based reward manager. Specifically, we employ `math-verify` for answer extraction and verification against the ground truth. We do not include a KL-regularization term in either the loss or the reward. The PPO update mini-batch size is the same as the training batch size, *i.e.*, , training is conducted in an on-policy manner.

Qwen3-1.7B and Qwen3-4B are trained under the thinking mode. Qwen2.5-VL-7B-Instruct does not natively possess strong complex mathematical reasoning or long-chain-of-thought capabilities, so its RL training starts from a checkpoint obtained after SFT on the AceReason dataset. For both training and evaluation, all models use the following system prompt and the following chat template:

---

**System Prompt**

You FIRST think about the reasoning process as an internal monologue and then provide the final answer. The reasoning process MUST BE enclosed within `<think> </think>` tags. The final answer MUST BE put in `\boxed{}`.

---

**Chat Template**

{question}
Please reason step by step, and put your final answer within \boxed{}.

---

### A.2. Speculative Decoding Pseudocode

For completeness, we provide the full BubbleSpec rollout procedure in Algorithm 1, including draft retrieval from the suffix tree, sequential verification under the current policy, acceptance/rejection handling, and fallback decoding after the first rejection. This pseudocode clarifies that BubbleSpec uses the retrieved continuation as a deterministic speculative proposal, while preserving the exact target rollout distribution through token-wise acceptance and residual resampling.

### A.3. Analysis of Utilizing Intra-GPU Bubbles

*Table 9.* Comparison of BubbleSpec using inter-gpu vs. intra-gpu bubbles.

| Method | Rollout Time (s) | Rollout Bubble Time (s) | | Draft Response Length | |
|---|---|---|---|---|---|
| | | **Average** | **Maximum** | **Average** | **Maximum** |
| BubbleSpec w/. Inter-GPU Bubble | 167.6 | 61.9 | 109.1 | 4169 | 24029 |
| BubbleSpec w/. Intra-GPU Bubble | 232.8 | 89.1 | 162.5 | 4524 | 26067 |

We analyze the effect of utilizing intra-GPU bubbles in this section using Qwen3-1.7B. Specifically, when the active batch size is below the threshold of 8, we exploit intra-GPU bubbles on 6 out of 8 ranks, in the order in which their active batch size falls below 8. As shown in Table 9, compared with utilizing only inter-GPU bubbles, additionally utilizing intra-GPU bubbles allows us to make better use of otherwise idle GPU time. However, this does not translate into a direct increase in

---

**Algorithm 1** BubbleSpec Rollout with Deterministic Draft Verification

---

**Require:** Prompt $x_{1:m}$, target policy $\pi$, suffix tree $\mathcal{T}$, maximum generation length $L$
**Ensure:** Rollout $y$
1: $y \leftarrow x_{1:m}$
2: **while** $|y| < L$ and not EOS **do**
3:     Retrieve a draft block $\tilde{d} = (\tilde{x}_1, \ldots, \tilde{x}_K)$ from $\mathcal{T}$ using prefix $y$
4:     **if** $\tilde{d}$ is empty **then**
5:         Sample $x \sim \pi(\cdot \mid y)$
6:         $y \leftarrow y \circ x$
7:         **continue**
8:     **end if**
9:     $acceptedAll \leftarrow$ **true**
10:     **for** $t = 1$ to $K$ **do**
11:         Let $p_t(\cdot)$ be the target decoding distribution under prefix $y$
12:         $a \leftarrow \tilde{x}_t$
13:         Accept $a$ with probability $p_t(a)$
14:         **if** accepted **then**
15:             $y \leftarrow y \circ a$
16:             **if** $a$ is EOS **then**
17:                 **break**
18:             **end if**
19:         **else**
20:             Sample $x \sim r_t(\cdot)$, where

$$r_t(x) = \frac{p_t(x)\mathbf{1}[x \neq a]}{1 - p_t(a)}$$

21:             $y \leftarrow y \circ x$
22:             $acceptedAll \leftarrow$ **false**
23:             **break**
24:         **end if**
25:     **end for**
26:     **if** $acceptedAll$ and last token is not EOS **then**
27:         **continue** {retrieve a new draft block from the extended prefix}
28:     **end if**
29: **end while**
30: **return** $y$

---

*Table 10.* BubbleSpec performance on Qwen3-1.7B across different RL algorithms.

| Method | Rollout Time (s) | Decoding Steps | | Response Length | | Speculative Metrics | | |
|--------|------------------|---------|---------|---------|---------|---------------|--------------|-------------|
| | | Average | Maximum | Average | Maximum | Accept. Length | Draft Length | Accept. Rate |
| SAPO | 167.6 | 2913 | 15908 | 6592 | 39685 | 2.15 | 3.84 | 29.4% |
| GRPO | 176.3 | 3032 | 17297 | 6844 | 39189 | 2.13 | 3.83 | 29.5% |
| GSPO | 174.4 | 3001 | 17742 | 6773 | 39580 | 2.14 | 3.83 | 29.8% |

the draft response length. Therefore, we argue that exploiting inter-GPU bubbles alone is sufficient to provide good coverage for next-step response generation.

In addition, we observe that utilizing intra-GPU bubbles leads to significantly longer rollout time. This is due to interference with the current batch's generation: adding next-batch prompts increases the effective batch size and slows down the decoding of the current batch. To illustrate this, we record the latest finish time among ranks that utilize inter-GPU bubbles and ranks that do not, respectively, as shown in Figure 7. We find that ranks that are faster in earlier generation stages can actually complete later than initially slower ranks, due to both the unpredictable LLM output length and intra-GPU interference. We argue that, to efficiently utilize intra-GPU bubbles without slowing down the current batch, techniques such as intra-GPU sharing and isolation need to be employed (Xu et al., 2025). Since the LLM decoding stage is memory-bound, this isolation should target not only compute resources but, more importantly, the GPU main memory bandwidth.

## A.4. Experiments on More RL Algorithms

We further evaluate BubbleSpec under additional RL algorithms in Table 10, including GRPO and GSPO, and observe broadly similar performance across algorithms. As a rollout optimization framework, BubbleSpec can be seamlessly

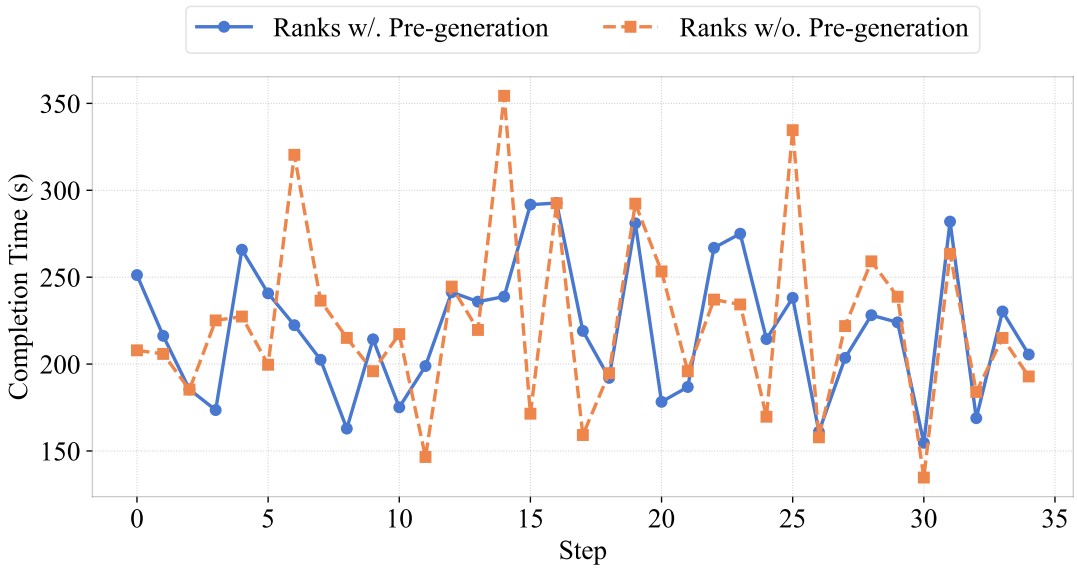

*Figure 7.* Latest completion time among ranks with and without response pre-generation.

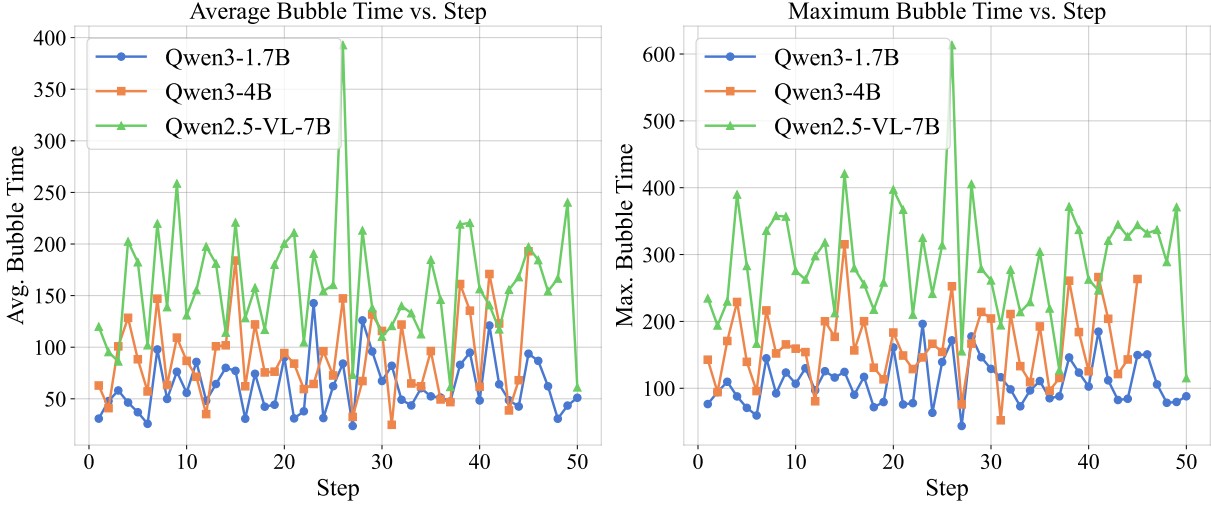

*Figure 8.* Average and maximum bubble time during training steps.

integrated into a wide range of RL algorithms.

## A.5. Bubble Time in RL Training

We report the average and maximum bubble times across DP ranks throughout training for the three models in Figure 8. Both metrics remain substantial relative to the total rollout latency, providing ample time for draft pre-generation and ensuring sufficient diversity and length coverage of speculative drafts.

### A.6. Test Accuracy on Qwen3-1.7B

The AIME25 test accuracy of Qwen3-1.7B after 300 SAPO training steps is reported in Table 11, and closely matches that obtained with vanilla Verl training without speculative rollout acceleration.

*Table 11.* Test accuracy on AIME25 of Qwen3-1.7B after SAPO training for 300 steps.

| Model | Qwen3-1.7B | Qwen3-1.7B (+BubbleSpec 300 steps) | Qwen3-1.7B (+Verl 300 steps) |
|---|---|---|---|
| Score | 36.8 | 43.3 | 41.6 |

