# OpenReview forum: "BubbleSpec: Turning Long-Tail Bubbles into Speculative Rollout Drafts for Synchronous Reinforcement Learning"
_ICML.cc/2026/Conference — ICML 2026 regular_

### Official Review · Reviewer_5Np6 · 2026-03-12

**Soundness:** 3
**Presentation:** 3
**Significance:** 3
**Originality:** 2
**Overall Recommendation:** 4
**Confidence:** 3

**Summary:**

This paper is proposing **BubbleSpec**, a novel framework that accelerates RL rollouts while strictly keeping the mathematical exactness. Instead of attempting to eliminate bubbles, BubbleSpec exploits them. It exploits the idle time windows of faster ranks to pre-generate rollout results for subsequent steps, serving as drafts for speculative decoding.

Extensive evaluations demonstrate that BubbleSpec reduces decoding steps by ~50% and increases rollout throughput by up to 1.8x. Critically, BubbleSpec is seamlessly compatible with various RL frameworks and strategies as it sustains the strict synchronous property of RL algorithms.

**Compliance With Llm Reviewing Policy:**

Affirmed.

**Final Justification:**

Most of my concerns are addressed. I am in favor of the idea. It's simple but effective.

**Key Questions For Authors:**

1. Is there any results comparison with other speculative sampling methods in RL, such *SpecRL*, particularly in the epocs later than the first one. The authors have discussed them in the related works, and claim that they've suffered from the code-start problem. I agree with this discussion, but I want to understand what's the quantitive gain over them after the cold-start phase, in terms of comparison of averaged accepted length and end-to-end training or rollout speed.

2. is this method highly tied to DP? for large models, other parallelism such as TP are widely deployed in the rollout phase. however, this may affect the bubbles, or maybe turn all inter-GPU bubbles into intra-GPU bubbles in extreme cases. Have the authors considered such cases and is there any exploration about it?

**Limitations:**

The authors have discussed the limitation and potential solutions in the paper and appendix, particularly about the intra-gpu bubbles.

**Strengths And Weaknesses:**

The paper is targeting one the fundamental challenges of RL infrastructure for LLMs, the long-tail problem in the rollout phase. Instead of exploring methods like async-RL to leverage the caused off-policy data, or eliminate the long-tail by batch reordering, this work smartly make use of the idle window by draft generation for speculative sampling in upcoming data.

To make it effectively work, the authors designed the unified attention mechanism to avoid slowing down the normal generation by the generating and verification overhead. The experiments show this method is efficient in reducing the rollout latency without changing the data distribution.

I am generally in favor of this paper, and please refer to the "key questions" part for the weakness in my mind. I'd like to update my score if the concerns are well addressed.

---

> ### Author Rebuttal · Authors · 2026-03-29
>
> We thank the reviewer for the positive evaluation and insightful feedback. Below, we provide detailed responses to your comments.
>
> ### **Q1: Speculative RL baselines**
>
> **Compared with Rhyme-RL.** We compare BubbleSpec with the history-based baseline Rhyme-RL. It uses responses from the previous epoch as draft proposals, with other settings such as attention kernel implementation, kept the same for a fair comparison. Due to time constraints, we truncate the dataset so that a full epoch spans 50 steps, and report results averaged over the second epoch using Qwen3-1.7B.
>
> | Method     | Rollout Time (s) | Avg Decode Steps | Max Decode Steps | Avg. Accept Length |
> | -- | -- | -- | -- | -- |
> | BubbleSpec | 167.6            | 2913             | 15908            | 2.15               |
> | Rhyme-RL   | 173.3            | 2972             | 16516            | 2.14               |
>
> As shown in the table, BubbleSpec achieves performance comparable to drafting from full responses generated in the previous epoch. First, history-based methods suffer from cross-epoch distribution shift, offsetting the advantage of longer historical responses. Second, we find that BubbleSpec’s partial responses are already sufficient to capture the diversity needed for next-batch rollout drafting, so the benefit of longer historical responses is marginal. Third, drafting from a much larger suffix tree adds additional overhead.
>
> **Compared with SpecRL.** SpecRL drafts from responses from prior epochs and accepts the longest prefix that the current policy can accept. It uses a lenient form of rejection sampling, which does not preserve a lossless rollout distribution. Following their relaxed rejection sampling, we report the average reduction in decoding steps under different lenience levels $l$. Due to time constraints, we did not run end-to-end experiments to measure rollout speed; however, decoding-step reduction serves as a reasonable proxy. The experiments are conducted on Qwen2.5-VL-7B-Instruct checkpoints on Polaris-53k.
>
> | Method                  | BubbleSpec | SpecRL (no lenience) | SpecRL ($l=e^{0.2}$) | SpecRL ($l=e^{0.5}$) | SpecRL ($l=e^{0.8}$) |
> | -- | --- | -- | -- | --- | --- |
> | Decoding Step Reduction | 4647       | 73                   | 272                  | 1175                 | 3375                 |
>
> SpecRL's average decoding-step reduction is smaller than that of BubbleSpec. In fact, SpecRL is evaluated in short-context settings, using a maximum response length of 4k in their paper. In long-context RL training, however, SpecRL can accept only prefixes from historical responses, so its relative speedup quickly diminishes as responses grow longer. Therefore, SpecRL is better suited to short-context scenarios, where long-tail bubbles are less pronounced.
>
> ### **Q2: In the case of larger TP sizes and fewer inter-GPU bubbles**
>
> We acknowledge that BubbleSpec, in its current form, primarily exploits inter-GPU bubbles and therefore relies on DP. We would like to provide two clarifications:
>
> **1. BubbleSpec can be extended to exploit intra-GPU bubbles with hardware partition techniques when TP size is large.**
>
> As discussed in Appendix A.2, we didn't exploit intra-GPU bubbles mainly because of intra-GPU resource contention. To ensure that newly added requests don't slow down rollout for the current batch, fine-grained request-level resource partitioning is needed to prioritize current-batch generation.
>
> Such support is enabled by GPU partitioning techniques like NVIDIA Green Context. Following prior work \[1], we have explored using these techniques to separate current-batch decoding from next-batch prefilling, which helps mitigate contention because decoding is memory-bound, whereas prefilling is more compute-bound. For decode–decode separation, however, additional memory-bandwidth partitioning would be required to effectively reduce interference. We view this as a promising direction for future work.
>
> **2. For larger models that require TP, a non-trivial DP degree still exists.**
>
> * TP degree is typically kept moderate, as larger TP increases inter-GPU communication overhead. As noted in Rollpacker Section 4.2 \[2], TP is enabled only when KV-cache capacity becomes a bottleneck. In practice, one typically uses the smallest TP degree that avoids severe KV-cache swapping.
>
> * The total GPU count generally grows with model size. As a result, even when TP is needed for larger models, a meaningful DP dimension often remains in real deployments.
> ---
> **Thanks for your time and consideration. We sincerely hope that you find our responses convincing and would consider increasing your rating.**
>
> References:
>
> [1] Xu, Yuhang, et al. "Nova: Real-Time Agentic Vision-Language Model Serving with Adaptive Cross-Stage Parallelization." 2025 IEEE Real-Time Systems Symposium (RTSS). IEEE, 2025.
>
> [2] Gao, Wei, et al. "Rollpacker: Mitigating long-tail rollouts for fast, synchronous rl post-training." arXiv preprint arXiv:2509.21009 (2025).

---

> > ### Author Rebuttal · Reviewer_5Np6 · 2026-04-03
> >
> > Thanks the authors for the informative rebuttal. I will keep my score since it seems that the proposed method achieve very similar throughput (token/s) compared with previous method (Rhyme-RL).
> >
> > I am willing to see more explanation and results about this.

---

> > > ### Author Response · Authors · 2026-04-04
> > >
> > > Thank you for the follow-up comment. First, we would like to clarify why BubbleSpec's performance is similar to Rhyme-RL in this comparison:
> > > 1. In our implementation, Rhyme-RL differs from BubbleSpec only in the source of suffix-tree construction, i.e., previous-epoch responses rather than partial rollouts, while the remaining system optimizations are matched for fairness. We acknowledges this is a simplified implementation of Rhyme-RL, as the original rhyme-RL is indeed an asynchronous RL framework that includes additional techniques beyond speculative decoding.
> > >
> > > 2. The current setting is favorable to Rhyme-RL. The reported results are measured in the second epoch after warm-up, and, due to time constraints, an epoch is shortened to 50 steps. This setup reduces inter-epoch response drift and therefore benefits history-based drafting. In more realistic settings with longer epochs, we expect the effectiveness of Rhyme-RL to degrade more noticeably.
> > >
> > > Through this comparison, we aim to highlight BubbleSpec’s competitiveness relative to history-based methods even after warm-up. In particular, the results show that, even without epoch warm-up, with short epochs, and using only partial rollouts, BubbleSpec achieves performance comparable to Rhyme-RL.
> > >
> > > BubbleSpec also avoids substantial storage overhead. Unlike history-based drafting, it does not require storing all responses from a full epoch. In our setting, the responses from a single step occupy roughly 100MB, which would accumulate to about 50GB over 500 steps—a considerable storage overhead.
> > >
> > > For these reasons, we believe BubbleSpec is more advantageous in long-epoch and long-context scenarios.
> > >
> > > -------
> > > Thanks for your time and consideration. We sincerely hope that our responses have addressed your concerns and clarified the value of our work.

---

### Official Review · Reviewer_kZK7 · 2026-03-12

**Soundness:** 2
**Presentation:** 3
**Significance:** 2
**Originality:** 2
**Overall Recommendation:** 4
**Confidence:** 4

**Summary:**

This paper studies rollout inefficiency in synchronous RL for LLM post-training. The core idea in BubbleSpec is to exploit idle GPU time caused by long-tail response lengths across data-parallel ranks: while slower ranks finish the current rollout step, faster ranks pre-generate partial rollouts for the next step, build per-prompt suffix trees over these drafts, and then use model-free speculative decoding to reduce future decoding steps. The paper also introduces a unified attention implementation to reduce speculative verification overhead. On Verl/vLLM with Qwen3-1.7B, Qwen3-4B, and Qwen2.5-VL-7B on 8 NVIDIA B200 GPUs, the authors report roughly 30%-45% rollout time reduction and 1.4x-1.8x rollout throughput gains relative to vanilla Verl, without obvious quality degradation.

**Compliance With Llm Reviewing Policy:**

Affirmed.

**Final Justification:**

Thank you for the detailed rebuttal. The response addresses my main concerns. In particular, the clarification that the suffix-tree draft should be viewed as a deterministic proposal with residual resampling makes the exactness claim much clearer, and the distinction between distribution-level equivalence and trajectory-level nondeterminism is helpful. The additional 400-step end-to-end results, 4-node experiments, and comparisons to prior speculative RL baselines also substantially strengthen the empirical case. I still think the paper would benefit from making the rejection-sampling procedure and exactness claim more explicit in the main text, but overall my concerns have been adequately addressed, and I have adjusted my score accordingly.

**Key Questions For Authors:**

Please provide a formal proof or a much more explicit derivation of the losslessness claim. What is the exact proposal distribution for draft tokens under suffix-tree retrieval, and how is it accounted for during acceptance/rejection?

What is the end-to-end wall-clock training speedup over a full RL run, including rollout, reward computation, synchronization, and policy updates, rather than rollout-only timing over a short horizon?

How does BubbleSpec scale beyond a single 8-GPU node? In particular, what happens to the centralized coordinator and synchronization overhead under multi-node training?

How competitive is BubbleSpec against prior speculative rollout baselines after those methods have completed their warm-up phase, or in hybrid settings where historical drafts are available?

How robust are the gains when sequence lengths are less skewed, bubble time is smaller, or tensor-parallel size is larger?

**Limitations:**

No. The paper should explicitly discuss at least the dependence on substantial inter-rank bubble time, the single-node and centralized-coordinator assumptions, the extra CPU/GPU memory and indexing overhead, the settings in which speculative verification may erase the gains, and the possible societal impact of reducing the cost of scaling reasoning-oriented RL.

**Strengths And Weaknesses:**

Strengths

The paper targets an important practical bottleneck. Rollout latency is a serious issue in synchronous long-context RL, and exploiting otherwise idle bubble time is a sensible direction.

The method is reasonably modular and preserves the synchronous structure of the RL pipeline rather than relying on explicitly stale/off-policy rollouts.

The empirical speedups are non-trivial, and the paper includes several useful ablations, including inter- vs. intra-GPU bubbles, suffix decoding vs. n-gram matching, unified attention, and transfer across several RL algorithms.

The paper is generally well organized and the high-level system idea is easy to follow.

Weaknesses

The strongest claim, namely that the method is mathematically exact/lossless, is not convincingly established. The paper proposes draft selection through suffix-tree matches over pre-generated responses, but the actual proposal distribution induced by this retrieval procedure is not clearly formalized. The acceptance rule is stated briefly using adjacent-step token probabilities, but this is not enough to make the exactness claim convincing without a fuller derivation.

The evaluation is too narrow for the ambition of the paper. Most efficiency numbers are averaged over only 50 steps, all experiments are on a single 8xB200 node, and there is no multi-node scaling study even though the paper positions itself as a large-scale RL systems contribution.

The paper mainly demonstrates rollout-time improvements rather than end-to-end RL training speedup. Since the full training loop also includes reward computation, synchronization, and optimization, the practical gain for complete training remains unclear.

Baseline coverage is limited. The paper compares mostly against vanilla Verl and RollPacker, but not against prior speculative rollout approaches such as history-based methods after warm-up, or model-based speculative baselines in comparable settings. This makes the empirical case less complete than it should be.

The overall novelty is moderate. The contribution is a thoughtful systems combination of pre-generation during bubbles, suffix-based retrieval, and attention-kernel optimization, but each ingredient is relatively close to existing ideas.

Accuracy preservation is only lightly validated. The main quality comparison is limited in scope, with little discussion of variance across runs or seeds.

The limitations and societal impact discussion is weak. The paper does not seriously discuss settings where bubble time is small, memory/indexing overheads, hardware/runtime dependencies, or the possibility that making reasoning RL cheaper can also make scaling potentially harmful models easier.

---

> ### Author Rebuttal · Authors · 2026-03-29
>
> We greatly appreciate the reviewer's detailed feedback and constructive questions. Below, we provide detailed responses to your comments.
>
> ### **Q1: Lossless Claim**
>
> Due to the word limit, we kindly refer you to those responses for a more detailed validation of exactness:
>
> Reviewer 1hL9 "A general clarification on rejection sampling", where we detail the token acceptance procedure and clarify possible misunderstandings.
>
> Reviewer Z5Jt "Q1: Lossless claim", where we justify the exactness in both theoretical and experimental aspects, with more results provided.
>
> ### **Q2: End-to-end speedup**
>
> First, we'd like to clarify that a full long-context RL run typically consists of hundreds of steps, with each step taking several minutes to tens of minutes. In practice, we terminate an RL run when no reward improvement is observed for a long period, which takes days to weeks. As our work focuses on system acceleration, we believe that evaluating 50 steps is sufficient to reflect actual speedups and stability in real-world scenarios.
>
> To further support this claim, we conduct additional experiments on Qwen3-1.7B for **400** steps. The full run takes \~60 hours with Verl and \~37 hours with BubbleSpec. We report both the end-to-end wall-clock time and a detailed breakdown, where all reported numbers are amortized over 400 steps.
>
> | Method     | E2E Time (s) | Rollout Time (s) | Actor Update (s) | Reward (s) | Weight Sync. (s) |
> | ---------- | ------------ | ---------------- | ---------------- | ---------- | ---------------- |
> | Verl       | 544          | 427              | 92               | 6.9        | 14.1             |
> | BubbleSpec | 337 (-38.1%) | 226 (-47.1%)     | 87               | 6.8        | 13.7             |
>
> It can be observed that rollout accounts for the majority of the end-to-end runtime (around 70%). Thus, BubbleSpec substantially accelerates the end-to-end training by reducing the rollout time.
>
> ### **Q3: Evaluation beyond a single node**
>
> As suggested, we conduct experiments on 4 nodes, each equipped with 8xB200 GPUs, using the Qwen3-VL-8B-Thinking model and Polaris-53k dataset. Due to the time and resource constraints, the efficiency results below are average across 80 training steps. It can be observed that BubbleSpec has considerable time reduction on both the rollout stage and end-to-end training in a larger-scale setup.
>
> | Method     | Rollout Time (s) | E2E Time (s) | Avg. Decode Steps | Max. Decode Steps | Avg. Resp. Length | Max. Resp. Length |
> | ---------- | ---------------- | ------------ | ----------------- | ----------------- | ----------------- | ----------------- |
> | Verl       | 587              | 855          | 15110             | 32000             | 15122             | 32000             |
> | BubbleSpec | 427 (-27.2%)     | 700 (-18.1%) | 8530 (-43.5%)     | 21023 (-34.3%)    | 15392             | 32000             |
>
> **Centralized Coordinator and Synchronization Overhead.** Verl adopts a single-controller architecture that has been shown to scale well in multi-node settings. Built on top of Verl, BubbleSpec introduces only minimal additional communication overhead, namely the transmission of partial responses for one extra batch. In a setting with batch size 128, 16 responses per prompt, a maximum response length of 32k, and 4 bytes per token, the extra data amounts to transmit is only $128 \times 16 \times 32k \times 4$ bytes, i.e., 256 MB. In practice, the overhead is much smaller, since average response lengths are typically far below the maximum. For modern AI training clusters with 100–400 Gbps inter-node bandwidth, this synchronization overhead is negligible.
>
> As for memory overhead, suffix-tree construction is distributed across rollout ranks. Consequently, in multi-node settings, the associated memory overhead is also distributed and remains stable as the number of nodes increases.
>
> ### **Q4: Speculative rollout baselines**
>
> Please refer to our response to Reviewer 5Np6 "Q1: Speculative RL baselines", for comparison with rhymeRL and SpecRL.
>
> ### **Q5: Less skewed sequence lengths, smaller bubble time, or larger tensor-parallel size**
>
> BubbleSpec targets real-world long-context RL training scenarios, which typically exhibit severe sequence-length skewness due to the randomness of LLM outputs. We acknowledge that the long-tail bubble problem may be less severe in short-context scenarios. Please find more details in our responses to Reviewer 5Np6, "Q2: In the case of larger TP sizes and fewer inter-GPU bubbles" for an extended clarification.
>
> ### **Other issues**
>
> We will add a discussion of the broader societal impacts and the application scenarios of BubbleSpec. Thank you for pointing this out！
>
> ***
>
> **Thank you for your time and consideration. We sincerely hope that you find our responses convincing and would consider increasing your rating.**

---

> > ### Author Rebuttal · Reviewer_kZK7 · 2026-04-04
> >
> > Thank you for the detailed rebuttal. The response addresses my main concerns. In particular, the clarification that the suffix-tree draft should be viewed as a deterministic proposal with residual resampling makes the exactness claim much clearer, and the distinction between distribution-level equivalence and trajectory-level nondeterminism is helpful. The additional 400-step end-to-end results, 4-node experiments, and comparisons to prior speculative RL baselines also substantially strengthen the empirical case. I still think the paper would benefit from making the rejection-sampling procedure and exactness claim more explicit in the main text, but overall my concerns have been adequately addressed, and I have adjusted my score accordingly.

---

> > > ### Author Response · Authors · 2026-04-04
> > >
> > > Thank you for taking the time to read our rebuttal and for your positive feedback. We agree that the rejection-sampling procedure and exactness claim should be made more explicit in the main text, and we will clarify this in the revision.

---

### Official Review · Reviewer_1hL9 · 2026-03-12

**Soundness:** 3
**Presentation:** 3
**Significance:** 3
**Originality:** 3
**Overall Recommendation:** 5
**Confidence:** 4

**Summary:**

This paper addresses suboptimal GPU utilization in RL-based LLM training caused by 'long-tail' responses. The authors propose leveraging pipeline bubbles to pre-generate partial rollouts for subsequent training steps using speculative decoding. Unlike previous methods that relax synchrony, this approach maintains mathematical exactness and losslessness while improving throughput by 1.8x.

**Compliance With Llm Reviewing Policy:**

Affirmed.

**Final Justification:**

I am updating my score from weak reject to accept based on the rebuttal clarifying several core concerns I had. After rebuttal 1. I believe the revision will have a more plausibly justified correctness over on-policy guarantee. 2. The algorithm specification is improved, the author has provided a more clear multi-token procedure. They commit to include pesudocode in the revised manuscript. 3. Sampling distribution is clarified appropriately. 4. The additional diagnostics in the 2nd round of rebuttal has made the performance story more coherent. They also commit to adding accepted-streak-length distribution, which is the right metric to support the throughput claims.

**Key Questions For Authors:**

1. At iteration n, is the suffix tree generated with the current policy $\pi_n$ every policy update step from scratch.
2. Do you align the proposal distribution p(x) with the $pi_n$ to make the equivalence.
3. Given your average response to construct the suffix tree, what is the average divergence between the proposal distribution q(x) and the actual current policy distribution $\pi_n(x)$?
4. When a draft token is rejected, do you sample the correction token from the residual distribution and restart the drafting? What's the rejecting ratio and will the GPUs wait on the draft rejecting and resampling instead, what is the impact of this?

**Limitations:**

If the method relies on speculative decoding to avoid distribution mismatch, state the exact conditions required (proposal distribution definition, correction sampling) and note what happens if they are violated.

**Strengths And Weaknesses:**

Strength:
1. The issue that the paper is trying to address is real and impactful.
2. The core idea is interesting and, at a high level, theoretically plausible.
3. The empirical results are good.

weakness:
1. My biggest concern is the acceptance-probability formulation. Is the suffix tree rebuilt from scratch at every policy update using samples from the current policy $\pi_n$? In the diagram, I assume it is but I feel the authors need to emphasise this to make it more explicit. Maybe with a short Pseducode (didn't read appendix, if it is in appendix, move it to the main text).
2. The accept equation $P=min(1, \frac{\pi_{n+1}(x)}{\pi_n(x)})$ where they directly use $\pi_n{(x)}$, treating $\pi_n(x)$ as the proposal distribution q(x). Even if the suffix tree is rebuilt from scratch from rollouts generated under $\pi_n$, it's not strictly equivalent to sampling from $\pi_n$. It is an emperical approximation with the token appearing frequencies, unless there is a design to align the q(x) to $\pi_n(x)$.
3. Given this, the synchronous claim in the paper is over-stated. The distribution is still staled.

---

> ### Author Rebuttal · Authors · 2026-03-29
>
> We thank the reviewer for thoughtful comments. Below, we provide detailed responses to your comments.
>
> ### **General clarification on rejection sampling**
>
> Since Weaknesses 1, 2, and 3, as well as Questions 2, 3, and part of Question 4, are all related to the token acceptance process, we would like to provide a general clarification here.
>
> BubbleSpec is a model-free speculative decoding method. In particular, we do not explicitly construct a draft distribution $q_t(x)$ from the suffix tree. Instead, at decoding step $t$, the suffix tree deterministically returns a drafted token $\tilde{x}_t$. This can be written as a degenerate proposal distribution $q_t(x)$, so that $q_t(\tilde{x}_t) = 1$, i.e., the deterministic distribution concentrated on the drafted token.
>
> Given the target model distribution $p_t(x)$, our implementation accepts the drafted token with probability
>
> $$
> \min\left(1,\frac{p_t(\tilde{x}_t)}{q_t(\tilde{x}_t)}\right)
> =\min(1,p_t(\tilde{x}_t))
> =p_t(\tilde{x}_t).
> $$
>
> Equivalently, this is a special case of rejection sampling with deterministic proposal $q_t(\tilde{x}_t)=1$.
>
> If the drafted token is rejected, we sample from the residual distribution obtained by masking out $\tilde{x}_t$:
>
> $$
> r_t(x)=\frac{p_t(x)\mathbf{1}[x\neq \tilde{x}_t]}{1-p_t(\tilde{x}_t)}.
> $$
>
> Therefore, the final output token is distributed as:
>
> $$
> \Pr(x_t=x) = p_t(x_t)\mathbf{1}[x=\tilde{x}_t] + (1-p_t(\tilde{x}_t))r_t(x) = p_t(x),
> $$
>
> which shows that the overall procedure exactly preserves the target distribution and hence the on-policy property.
>
> At the same time, we acknowledge that tokens retrieved from the suffix tree are not identical to samples drawn from the previous-step policy distribution; rather, they should be viewed as an empirical heuristic. In this sense, the current wording around Eq. (2) may be misleading if it suggests that the suffix-tree draft is itself sampled from a policy distribution. We will revise Eq. (2) and the surrounding text to make clear that the suffix tree provides a deterministic proposal.
>
> ### **Question 1**
>
> Yes, we rebuilt the suffix tree from scratch using the previous step's generated responses of the current batch.&#x20;
>
> ### **The Latter Part of Question 4**
>
> We reported the acceptance ratio of drafted tokens in the paper. Accordingly, here we provide the token rejection ratio of BubbleSpec:
>
> | Model           | Qwen3-1.7B | Qwen3-4B | Qwen2.5-VL-7B |
> | --------------- | ---------- | -------- | ------------- |
> | Rejection Ratio | 70.1%      | 71.4%    | 78.3%         |
>
> **Impact of rejection sampling on GPU idleness:** The rejection sampling procedure is now performed on the GPU using Triton kernels. However, it still remains on the critical path of LLM decoding. We profile the decoding step using `torch.profiler` and report the corresponding wall-time overhead on both the CPU and GPU sides, as shown below.
>
> $\ $     | Single Step | Rejection Sampling | Overhead Ratio
> -----|-----|--------|----
>  CPU side | 18.3 ms                   | 1.67 ms            | 9.1%
>  GPU side | 13.4 ms                   | 0.417 ms           | 3.1%
>
> We observe that the CPU-side overhead is substantially higher than the GPU-side overhead, thus leading to GPU underutilization. To alleviate this issue, techniques like asynchronous scheduling with speculative decoding have been proposed. By overlapping CPU-side scheduling and bookkeeping with GPU execution, this design reduces GPU idle time and improves overall throughput. Currently, BubbleSpec does not employ asynchronous scheduling due to its lack of support in vLLM, but theoretically, it can be incorporated to further improve throughput.
>
> ### **Other issues**
>
> We will revise the paper to make both the rejection sampling procedure and the acceptance-probability formulation clear. Thank you for pointing this out!
>
> ***
>
> **Thank you for your time and consideration. We sincerely hope that you find our responses convincing and would consider increasing your rating.**

---

> > ### Author Rebuttal · Reviewer_1hL9 · 2026-04-03
> >
> > Thanks for the detailed clarification — it helps, and it also confirms that my original concern was not just a presentation issue.
> >
> > 1. **Eq. (2) / proposal distribution is materially misleading as written.**
> >    You now describe BubbleSpec as rejection sampling with a *deterministic* proposal $q_t$ concentrated on the suffix-tree token $\tilde{x}_t$ (i.e., $q_t(\tilde{x}_t)=1$), with accept probability $p_t(\tilde{x}_t)$ and residual sampling over $x\neq \tilde{x}_t$. This is a different story than Eq. (2) suggesting $q=\pi_n$ and acceptance $\min(1,\pi{n+1}/\pi_n)$. Please ensure the main text is revised accordingly, including the motivation: acceptance depends on the *quality* of the suffix-tree proposal under the current policy, not primarily on consecutive-policy closeness.
> >
> > 2. **Please provide complete multi-token pseudocode (critical for “lossless / on-policy” claims).**
> >    The rebuttal gives a correct single-step argument. However, the paper’s core runtime uses *multi-token* speculative decoding. For the “exactly preserves the target distribution / on-policy” claim to be convincing, the main paper should include pseudocode for the full procedure: accept drafted tokens sequentially until the first rejection; on rejection, sample from the residual distribution at that position; and then restart drafting from the updated prefix (and clarify whether you sample a “bonus” token when all drafted tokens are accepted). Without this, it’s hard to assess whether the implemented algorithm matches the claimed distributional guarantee.
> >
> > 3. **Clarify what $p_t$ is in practice (temperature/top-p/top-k).**
> >    The rejection sampling guarantee applies to the *actual* sampling distribution used at decode time. Please explicitly state whether $p_t$ refers to raw softmax probabilities or the post-processed distribution after temperature/top-p/top-k (and confirm that the residual sampling uses the same $p_t$.
> >
> > 4. **High rejection rates raise a new consistency question with the reported speedups.**
> >    You report rejection ratios of ~70–78%. That implies the suffix-tree deterministic token has low acceptance most of the time. This does not invalidate correctness, but it makes the performance story less intuitive. To reconcile this with the large decoding-step / throughput improvements, it would help to report: (i) accepted streak length distribution (not just token-level acceptance), (ii) expected $p_t(\tilde{x}_t)$ and/or rank of $\tilde{x}_t$ under $p_t$, and (iii) a wall-time breakdown of “verify + rejection sampling + restart” overhead vs. savings. Right now, the narrative “policies are close ⇒ high acceptance” seems inconsistent with the rejection rates, and should be updated.
> >
> > 5. **Tree rebuild timing remains slightly ambiguous.**
> >    You state the tree is rebuilt “from scratch using the previous step’s generated responses of the current batch.” Please spell out timing precisely relative to (a) the policy update and (b) the next batch / prompt set. This matters both for practicality (how much bubble time is needed) and for proposal quality/cold-start behavior.
> >
> > Overall: I like the idea. The deterministic-proposal rejection sampling framing makes the *on-policy* claim more plausible, but it also means the paper needs to (i) correct Eq. (2) and its surrounding interpretation, and (ii) add the missing algorithmic specification and diagnostics so readers can evaluate both correctness and why the method is fast despite high rejection. If the concerns can addressed in the following response, I will increase rating.

---

> > > ### Author Response · Authors · 2026-04-03
> > >
> > > ### **Ack1: revision arounding Eq (2)**
> > > Thank you for pointing this out. We agree that the current presentation around Eq. (2) can be misleading and should be revised to better match the actual procedure. We will also clarify the motivation: the acceptance rate is primarily determined by how well the suffix-tree proposal matches the current policy distribution, rather than mainly by the closeness between consecutive policies.
> > >
> > > ### **Ack2: multi-token procedure**
> > > We agree that the full multi-token procedure should be described explicitly in the main paper. Below, we provide the detailed multi-token acceptance process. In the revised manuscript, we will include formal LaTeX pseudocode to present the complete procedure:
> > >
> > > * at each decoding step, retrieve a block of draft tokens from the suffix tree under the current prefix; run the target model over this draft block to obtain the target decoding distributions;
> > > * process the drafted tokens **sequentially**, applying rejection sampling at each position to decide whether the drafted token is accepted;
> > > * upon the **first rejection**, sample one recovered token from the corresponding residual distribution and stop processing the remaining drafted suffix;
> > > * if all drafted tokens in the block are accepted, additionally sample one **bonus token** from the target distribution;
> > > * update the decoded sequence with the accepted / recovered / bonus tokens; use the updated prefix to retrieve the next draft block from the suffix tree and continue decoding.
> > >
> > >
> > > ### **Ack3: what $p_t$ is in practice**
> > > In our implementation, $p_t$ is not the raw softmax over the model logits. Rather, $p_t$ denotes the post-processed target sampling distribution after all decode-time transformations have been applied. This includes temperature scaling and, when enabled, top-p and top-k filtering (we use temperature 1.0 in the experiments as reported in the paper).
> > >
> > > Both the acceptance probability and the residual correction step are defined with respect to this same final decoding distribution, rather than the raw logit softmax.
> > >
> > > As suggested, we will make this explicit in the revised method description and pseudocode.
> > >
> > > ### **Ack4: High rejection rates vs. large speedups**
> > > We believe there may be a small misunderstanding in how the reported rejection rate should be interpreted.
> > >
> > > Our reported rejection rate is simply $1 -$ accept rate, and the accept rate is computed only over drafted tokens. In particular, it does not include the recovered/bonus token. As described in our experimental setup, we draft 4 tokens from the suffix tree at each step. Therefore, an acceptance rate of roughly 25% means that, on average, about one drafted token is accepted. After accounting for the recovered token or the bonus token, each speculative step yields about two newly generated tokens on average. This is consistent with the roughly 50% decoding-step reduction reported in the paper.
> > >
> > > We agree, however, that the current narrative is incomplete, and that token-level acceptance statistics alone are not sufficient to fully explain the speedup. In particular, accepted streak length distribution is an important metric. Our current experimental scripts only log the average accepted length, but in the revised version we will add the full accepted-streak-length distribution as well.
> > >
> > > Regarding your phrase "verify + rejection sampling + restart" overhead, we interpret “verify” as the target-model forward pass. We are not fully sure what "restart" referred to in this context, but we assume it corresponds to the overhead of retrieving the next draft from the suffix tree; if so, the table below provides the wall-clock-time breakdown of a single step. Here is an anonymous URL to the image of the profiling trace: [url](https://anonymous.4open.science/r/ICML_Rebuttal-CEBF/overhead.png)
> > >
> > > Model Forward    | Rejection Sampling | Draft Token Retrieval | Others (Scheduling, etc.)
> > > -------|----|---|---
> > >  13.5 ms | 1.6 ms  | 1.3 ms | 1.7 ms
> > >
> > > Due to the overhead of rejection sampling and draft-token retrieval, the speedup is smaller than the reduction in decoding steps, as reported in the paper.
> > >
> > > ### **Ack5: tree construction timing**
> > > We agree that the timing of tree rebuilding should be stated more precisely. In our pipeline, the order is:
> > >
> > > 1.  current-batch rollout (during which we also generate the partial rollouts for the next batch),
> > > 2. current-batch policy update,
> > > 3. suffix-tree construction from scratch,
> > > 4. next-batch rollout using the rebuilt suffix tree.
> > >
> > > Therefore, tree construction is on the critical path rather than hidden in rollout bubbles. However, this overhead is quite lightweight in practice, as shown in Table 4 in the paper.
> > >
> > > ---
> > >
> > > We sincerely appreciate your time and effort in helping us improve this work.

---

### Official Review · Reviewer_Z5Jt · 2026-03-13

**Soundness:** 3
**Presentation:** 3
**Significance:** 3
**Originality:** 3
**Overall Recommendation:** 5
**Confidence:** 3

**Summary:**

This paper proposes BubbleSpec, a synchronous RL rollout acceleration method that converts inter-GPU rollout bubbles into speculative drafts for the next step. The method combines cross-step pre-generation, suffix-tree-based draft retrieval, and a unified attention implementation. Experiments show clear reductions in decoding steps and rollout time with similar final accuracy to a vanilla synchronous baseline.

**Compliance With Llm Reviewing Policy:**

Affirmed.

**Final Justification:**

I have no more questions.

**Key Questions For Authors:**

1. Please also clarify whether the exactness claim is distribution-level or trajectory-level, and provide stronger evidence on fairness/stability if possible.
2. The method is motivated as a synchrony-preserving alternative to asynchronous RL. Could the authors comment on whether BubbleSpec is also applicable in asynchronous or off-policy settings, and if so, whether the same exactness and stability arguments would still hold?

**Limitations:**

No. The paper does a reasonable job on technical limitations, but the discussion of potential negative societal impact is quite limited. It would be improved by briefly noting that faster RL training can lower the cost of scaling post-training for powerful LLMs, which may accelerate misuse, increase total compute consumption despite per-run efficiency gains, and widen access gaps between groups with and without large-scale infrastructure.

**Strengths And Weaknesses:**

Strengths:
The paper addresses an important systems bottleneck in RL for LLMs. The core idea is intuitive and practically useful: exploit bubble time rather than move to asynchronous RL. The combination of speculative drafting and unified attention is meaningful, and the empirical speedups are substantial. The paper is also well written, and the ablations are generally informative.

Weaknesses:
1. My main concern is the strong “lossless” / “exactness” claim. The paper does not clearly specify the exact speculative acceptance/rejection procedure for model-free suffix drafts; in particular, Eq. (2) seems to depend on the previous-step policy, but it is unclear how this quantity is obtained in practice. Relatedly, I am not fully convinced about experimental fairness: for LLM inference, even small backend or kernel changes can alter logits and sampled trajectories, and BubbleSpec changes both the speculative execution path and the attention implementation. Thus, the paper should clarify whether the claim is distributional equivalence in theory, rather than exact implementation-level equivalence.
2. The empirical scope is also somewhat limited: evaluation is single-node, and comparison to prior speculative RL baselines is mostly qualitative.

Overall, I think this is a good paper. I will raise my score if my concern is solved.

---

> ### Author Rebuttal · Authors · 2026-03-29
>
> We sincerely thank the reviewer for the positive evaluation and insightful feedback. Below, we provide detailed responses to your comments.
>
> ### **Q1: Lossless claim**
>
> **Theoretical Perspective:** BubbleSpec uses the same rejection-sampling mechanism as standard model-free speculative decoding; the difference lies in how draft tokens are proposed. Therefore, its rollout distribution is identical to that of decoding without speculative decoding. Since RL training relies on non-greedy sampling, exact trajectory-level equivalence is neither guaranteed nor practically attainable due to the inherent randomness of sampling. Even with a fixed random seed, LLM generation is known to be nondeterministic due to the floating-point effects. More details on the token acceptance procedure and clarifications on potential misunderstandings can be found in our response to Reviewer 1hL9, “A general clarification on rejection sampling”.
>
> **Experimental Perspective:** We acknowledge that kernel-level implementations can affect logit precision, leading to training–inference mismatches in RL. Since BubbleSpec introduces new kernels for unified attention, even if the rollout distributions are theoretically equivalent, this does not necessarily imply exact end-to-end performance equivalence.
>
> To demonstrate practical fairness, we report intermediate reward changes for training Qwen2.5-VL-7B on Polaris-53k, providing a strong indicator of RL training stability. As shown, BubbleSpec and Verl exhibit nearly identical upward reward trends.
>
> | Step       | 10    | 20    | 30    | 40    | 50    | 60    | 70    | 80    |
> | ---------- | ----- | ----- | ----- | ----- | ----- | ----- | ----- | ----- |
> | BubbleSpec | 0.705 | 0.716 | 0.727 | 0.736 | 0.748 | 0.757 | 0.760 | 0.766 |
> | Verl       | 0.704 | 0.715 | 0.726 | 0.736 | 0.747 | 0.757 | 0.759 | 0.766 |
>
> In addition, we report the average Pearson chi-square divergence between the training and rollout distributions to directly measure whether BubbleSpec exacerbates the training–inference mismatch at runtime.
>
>  Method             | BubbleSpec | Verl  |
> ------|----|---------|
>  $\chi^2$ divergence | 0.0083     | 0.0087        |
>
> Notably, since training–inference mismatch is unavoidable, a common way to mitigate its effect on training stability is to apply token- or sequence-level gradient masking to preserve the on-policy properties of the samples. BubbleSpec follows the same strategy. We refer the reviewer to the Verl Rollout Correction documentation for more details.
>
> ### **Q2: Extension to asynchronous or off-policy scenarios**
>
> We'd like to answer this question in two cases:
>
> 1. **Off-policy but still synchronous:** A common example is performing multiple updates from a single rollout batch. In this setting, BubbleSpec remains fully applicable, and the same exactness and stability properties hold. The key point is that the rollout process itself remains unchanged; only the collected data are reused or split into mini-batches for multiple updates.
>
> 2. **Asynchronous:** In asynchronous RL training, the rollout and training engines are disaggregated. The rollout engine continues generating samples and only stops when weight synchronization is triggered or the staleness threshold is reached. In this setup, asynchronous training eliminates the bubbles that arise in synchronous training. Consequently, BubbleSpec is no longer applicable, as it is specifically designed to exploit those idle periods.
>
> In addition, we'd like to emphasize that, in industrial practice (as reported for Kimi K2.5 and GLM5), synchronous on-policy training is commonly adopted for general or reasoning-oriented RL, where trajectory freshness is more critical. Instead, asynchronous training is often preferred for agentic RL to improve efficiency, as it can tolerate a higher degree of off-policyness. BubbleSpec is specifically designed for the former setting.
>
> ### **More evaluations**
>
> We refer you to the following responses for more evaluations:
>
> **Beyond single node:** Reviewer kZK7 "Q3: Evaluation beyond a single node", for results on 4 nodes each equipped with 8xB200s.
>
> **Comparison with prior speculative RL baselines:** Reviewer 5Np6 "Q1: Speculative RL baselines", for comparison with rhymeRL and SpecRL.
>
> ### **Limitations**
>
> We will include a more detailed discussion of the societal impacts of faster RL training and the application scenarios of BubbleSpec. Thank you for pointing this out！
>
> ---
>
> **Thanks for your time and consideration. We sincerely hope that you find our responses convincing and would consider increasing your rating.**

---

> > ### Author Rebuttal · Reviewer_Z5Jt · 2026-04-05
> >
> > I will raise my score.

---

### Decision · Program_Chairs · 2026-04-30

**Decision:**

Accept (regular)

**Comment:**

This submission studies the bottleneck in synchronous RL for LLM post-training. Its main idea is to use idle GPU time caused by long-tail rollout imbalance to pre-generate draft tokens for later speculative decoding, while keeping the synchronous RL pipeline unchanged.

There is consensus among reviewers with two acceptance and two weak acceptance recommendation after the rebuttal stage. Reviewers agreed that the studied problem is important, the proposed idea is practical, and the reported empirical speedups are strong. The rebuttal clarified the exactness claim and added more results and comparisons with prior speculative methods.

The strengths identified by reviewers:
- 1.	It studies an important problem and proposes a practical idea. All reviewers agreed that rollout bubbles are a real bottleneck in synchronous RL, and that using bubble time instead of moving to async RL is a useful direction.
- 2.	Its empirical results show strong efficiency. The experiments show large rollout speedups, fewer decoding steps (Z5Jt, kZK7, 5Np6).

The weaknesses raised by reviewers:
- 1.	Some claims are overstated. Reviewers mentioned that  the acceptance rule and full multi-token procedure were not clearly explained enough in the submission, especially for a strong “lossless” claim (Z5Jt, 1hL9, kZK7).
- 2. Reviewers noted limited discussion of settings with smaller bubbles, dependence on DP, memory/runtime overhead, and broader societal impact of cheaper RL training (Z5Jt, kZK7, 5Np6).

Overall, the recommendation is accept, as the reviewers are overall positive, and the scores moved in a favorable direction after rebuttal. The submission addresses an important systems problem, proposes a simple and effective method, and shows empirical speedups.